# Cancer Screening Prevalence among Participants in the Southcentral Alaska Education and Research towards Health (EARTH) Study at Baseline and Follow-Up

**DOI:** 10.3390/ijerph20166596

**Published:** 2023-08-18

**Authors:** Lauren C. Smayda, Gretchen M. Day, Diana G. Redwood, Julie A. Beans, Vanessa Y. Hiratsuka, Sarah H. Nash, Kathryn R. Koller

**Affiliations:** 1Alaska Native Tribal Health Consortium, Anchorage, AK 99508, USA; gmday@anthc.org (G.M.D.); dredwood@anthc.org (D.G.R.); katchk8@gmail.com (K.R.K.); 2Southcentral Foundation, Anchorage, AK 99508, USA; jbeans@southcentralfoundation.com; 3Center for Human Development, College of Health, University of Alaska Anchorage, Anchorage, AK 99508, USA; vyhiratsuka@alaska.edu; 4Department of Epidemiology, College of Public Health, University of Iowa, Iowa City, IA 52242, USA; sarah-nash@uiowa.edu

**Keywords:** Alaska native, cancer, longitudinal, cohort study, colorectal cancer, breast cancer, cervical cancer, screening, prevention

## Abstract

Alaska Native communities are working to prevent cancer through increased cancer screening and early detection. We examined the prevalence of self-reported colorectal (CRC), cervical, and breast cancer screening among Alaska Native participants in the southcentral Alaska Education and Research toward Health (EARTH) study at baseline (2004–2006) and ten-year follow-up (2015–2017); participant characteristics associated with screening; and changes in screening prevalence over time. A total of 385 participants completed questionnaires at follow-up; 72% were women. Of those eligible for CRC screening, 53% of follow-up participants reported a CRC screening test within the past 5 years, significantly less than at baseline (70%) (*p* = 0.02). There was also a significant decline in cervical cancer screening between baseline and follow-up: 73% of women at follow-up vs. 90% at baseline reported screening within the past three years (*p* < 0.01). There was no significant difference in reported breast cancer screening between baseline (78%) and follow-up (77%). Colorectal and cervical cancer screening prevalence in an urban, southcentral Alaska Native cohort declined over 10 years of follow-up. Increased cancer screening and prevention are needed to decrease Alaska Native cancer-related morbidity and mortality.

## 1. Introduction

Tribal leadership and the Alaska Tribal Health System have been actively working to prevent cancer and detect it earlier among Alaska Native people through increased colorectal, cervical, and breast cancer screening. As a result of these efforts, cancer screening among Alaska Native people has increased since 1999 [1,2]. Alaska Behavioral Risk Factor Surveillance System data (2018) showed that Alaska Native cancer screening rates are equal to or higher than those of non-Alaska Native people [1].Yet cancer is still a leading cause of morbidity and mortality among Alaska Native people [3,4,5,6]. The incidence of cancers that are detectable by regular age-appropriate screening among Alaska Native people during the most recent five-year period is as follows: colorectal cancer (CRC; 87.6 per 100,000), cervical cancer (12.4 per 100,000), and breast cancer (130.8 per 100,000) (2014–2018) [6]. Alaska Native people experience a disproportionate burden of these cancers: Alaska Native CRC rates are the highest in the world, over 2-fold higher than U.S. whites [6,7], and Alaska Native women are at nearly twice the risk for cervical cancer as U.S. white women [6]. However, breast cancer incidence among Alaska Native women is similar to that among U.S. white women [1,6]. A continued focus on screening and early detection can help reduce the burden of this disease [8]. Reasons for high CRC and breast cancer incidence among Alaska Native people are not well understood and may be related to behavioral factors and additional environmental factors associated with higher risk or needing a longer time horizon to see improvements in morbidity and mortality resulting from the recent increases in cancer screening [3,7].

The Alaska Education and Research toward Health (EARTH) study was established to improve understanding of risk and protective factors for cancer and other chronic diseases and to inform the design of more effective primary and secondary prevention strategies among Alaska Native people [3,9,10]. The Alaska EARTH study originally enrolled Alaska Native study participants between 2004 and 2006. The baseline data included self-reported cancer screening among Alaska Native and American Indian people living in three Alaska regions: two rural (southwest and southeast Alaska) and one urban (southcentral Alaska). Cancer screening prevalence among the baseline cohort in 2004–2006 varied by education status, income level, and the presence or absence of chronic medical conditions [11]. Additionally, rural residents were less likely to have received age- and sex-appropriate cancer screening tests than urban residents. Individuals who spoke only English at home, compared to those who spoke their Native language, were more likely to have gotten a colonoscopy/sigmoidoscopy for CRC screening. Marital status was related to the use of Pap tests, with married participants more likely to have received a Pap test in the past three years [11].

During 2015–2017, all EARTH study participants living in the urban, southcentral Alaska region were invited to participate in a 10-year follow-up study visit. We report here on colorectal, cervical, and breast cancer screening prevalence in the EARTH study follow-up cohort compared with the baseline southcentral Alaska cohort. We further identify factors associated with cancer screening at follow-up compared with baseline and describe trends in screening prevalence over time.

## 2. Methods

### 2.1. Ethics

The Alaska Area Institutional Review Board and the Tribal research review committees for the Alaska Native Tribal Health Consortium and Southcentral Foundation approved this study and manuscript [12]. All EARTH study participants provided written informed consent.

The Alaska Tribal Health System is a dynamic and self-governing system administered by and for Alaska Native and American Indian people [13]. Within this system, tribes and tribal health organizations manage health care delivery and have ownership and oversight of health research conducted with Alaska Native people [12,13]. Health challenges experienced by Alaska Native and American Indian people compared with other U.S. populations have also spurred tribal leaders and organizations to engage in health research such as the Education and Research Towards Health (EARTH) study, a national cohort study designed to investigate cancer and chronic disease risk factors among Alaska Native and American Indian people [10,12].

### 2.2. Study Eligibility

The 2004–2006 baseline EARTH study methods are described in detail elsewhere [10]. Investigators used identical survey methods and measurement instruments at follow-up as in the original EARTH study [10,11,14]. Participants were eligible to enroll in the study if they were Alaska Native/American Indian, aged ≤18 years old, not pregnant or receiving cancer chemotherapy, and able to give informed consent to participate in the study visit. Pregnant women could participate in the study once they were three months post-partum, and people receiving chemotherapy could enroll once they were one-year post-treatment [14].

### 2.3. Data Collection and Variables

Participants completed the same questionnaires at baseline and at follow-up. The questionnaires included demographic information, personal medical history, personal tobacco use history, and family history of cancer. Demographic variables included marital status (married or living as married vs. all other categories), education level (greater than high school completion vs. high school completion or less), income level (≥$40,000 vs. <$40,000), chronic medical condition(s) (0 or 1 vs. 2 or more), use of cigarettes (100 or more ever vs. never) or smokeless tobacco (ever vs. never), and language spoken at home (Alaska Native and/or American Indian language vs. English/other).

Participants were asked about their history of CRC, cervical, and breast cancer screening; participants were not asked about lung cancer screening since this was not available in the Alaska Tribal Health System at the time of baseline data collection (Table 1). If a participant refused to answer a question or did not know their age at their last screening, they were considered ineligible and excluded from data analysis [10,11]. Men and women 50 years of age or older were considered to have had appropriate CRC screening if they reported having a colonoscopy or sigmoidoscopy in the past five years. Women 18 years of age or older who reported not having had a hysterectomy were eligible for cervical cancer screening questions. Women 18 years of age or older were considered to have been appropriately screened for cervical cancer if they had a Pap test in the last three years. Women were considered to have been appropriately screened for breast cancer if they were 40 years of age or older and had a mammogram in the past two years.

### 2.4. Statistical Analysis

We compared paired demographic, personal medical history, family health history, and tobacco use variables between baseline and follow-up using McNemar’s test. We calculated the proportion of people screened for each of the cancer screening tests at each time point by dividing the number of people screened within the appropriate time frame by the number of people eligible for the test. Pearson’s Chi-Square test with Yates’ correction was used to identify statistically significant differences in rates between baseline and follow-up. We investigated which demographic, personal medical history, family health history, and tobacco use variables were associated with screening at baseline and separately at follow-up for each screening test, using logistic regression and controlling for age and sex as appropriate. Odds ratios are reported with significance demonstrated by 95% confidence intervals (CIs) not including 1.0. For all analyses, statistical significance was defined as *p* < 0.05. We used the R Core Team *(2021)* to conduct descriptive and multivariable-adjusted analyses [15].

## 3. Results

### 3.1. Demographic, Personal Medical History, Family Health History, and Tobacco Use Characteristics

Of the 1320 original southcentral Alaska EARTH study participants, 637 had valid contact information. These participants were invited to participate in the follow-up study (Figure 1). Of these, 385 participants (61%) completed the follow-up study visit, comprising 29% of the original invited cohort. Demographic, personal medical history, family cancer history, and tobacco use characteristics are shown in Table 2. The mean age at baseline was 40.3 years, and it was 51.9 years at follow-up. The proportion of women stayed the same at 72% at baseline and follow-up. The majority of participants had more than a high school education at baseline (65%) and follow-up (68%). At baseline, 32% reported annual household incomes >$40,000, compared to 50% at follow-up. Being married/living as married was marginally less prevalent at baseline (46%) compared to follow-up (47%). While 14% of participants at baseline spoke their Native language at home, this increased to 18% at follow-up. At baseline, 64% reported two or more chronic medical conditions, compared to 88% at follow-up. Family history of any cancer was more prevalent at baseline than at follow-up (47% vs. 41%). However, a family history of breast cancer was less prevalent among baseline participants (10%) than follow-up participants (17%), as was a family history of CRC (20% at baseline vs. 32% at follow-up). Fewer participants reported cigarette use (65% vs. 67%) at baseline compared to follow-up. However, more participants reported using smokeless tobacco (19% vs. 16%) at baseline compared to follow-up.

Statistically significant differences in the cohort between baseline and follow-up include older age (*p* < 0.01) in the follow-up cohort; decreased reported family history of any cancer (*p* = 0.03); however, increased reported family history of breast (*p* = 0.02) and colorectal (*p* < 0.01) cancers; a greater proportion reporting one or more chronic medical conditions (*p* < 0.01); a greater percentage reporting incomes ≥$40,000; and an increased proportion reporting speaking a Native language at home (*p* = 0.03).

### 3.2. Screening Test Prevalence

Participants eligible to be screened and the proportion of those participants who received cancer screening tests (colonoscopy/sigmoidoscopy, Pap test, and mammogram) at baseline and follow-up are shown in Table 3. Of those eligible for CRC screening at baseline, 70% reported having had a colonoscopy/sigmoidoscopy within the past five years, compared to 53% reporting CRC screening at follow-up (*p* = 0.02). Notably, fewer participants reported having had CRC screening more than five years ago at baseline (8%) than at follow-up (28%), while more reported never being screened at baseline (22%) than at follow-up (18%). Factors associated with colonoscopy/sigmoidoscopy in the past five years are shown in Table 4. Being female was significantly associated with CRC screening at baseline (*p* < 0.01), and having a family history of CRC was significantly associated with CRC screening at follow-up (*p* = 0.03). No other characteristics examined were significantly associated with CRC screening at baseline or follow-up.

The number of women eligible for cervical cancer screening decreased from baseline (*n* = 244) to follow-up (*n* = 160), since considerably more women at follow-up answered that they didn’t know when they had last received a Pap test and thus had to be excluded from the analysis (Table 3). Among those who were able to give information on time since screening, there was an overall significant (*p* < 0.01) decline in cervical cancer screening within the past three years between baseline (90%) and follow-up (73%). However, more women at follow-up reported having had a Pap test more than three years ago (23%) than at baseline (8%) (*p* < 0.01). More women also reported never having a Pap test at follow-up (5%) than at baseline (2%), although this difference was not statistically significant. Factors associated with Pap testing among women 18+ years of age at baseline and at follow-up are shown in Table 5. At follow-up, women >40 years were significantly more likely to get screened than those ≤40 years (OR = 2.7, *p* = 0.01). No other factors were found to be significantly different from baseline to follow-up (Table 5).

A total of 150 women were eligible for a mammogram at baseline, compared to 176 at follow-up. Reported breast cancer screening was similar between baseline and follow-up (78% vs. 77%). Among the baseline cohort, women ≥50 years old were more likely to have had a mammogram than those aged 40–49 years (OR = 3.4; *p* < 0.01); however, this association was not significant at follow-up. No other factors were significantly associated with mammography at baseline or follow-up (Table 6).

## 4. Discussion

We examined the prevalence of colorectal, cervical, and breast cancer screening among an urban cohort of Alaska Native individuals assessed at baseline and over 10 years of follow-up. We found that cancer screening prevalence in this cohort did not increase. CRC screening significantly declined from baseline to follow-up and had the highest percentage of people who had never been screened and the lowest percentage of those screened both at baseline and follow-up among the three cancer screening tests. However, a family history of CRC was associated with an increased likelihood of CRC screening at follow-up. Helping CRC survivors inform their family members and encourage screening may help build on this positive finding to further increase screening prevalence. Because of the high rates of CRC among Alaska Native people [6,16], there has been an increased focus on CRC screening by tribal and clinical leadership within the Alaska Tribal Health System, including participation in the Centers for Disease Control and Prevention CRC Control Program in 2009–2015 and 2020–2025 [17,18]. The goal of the program is to increase screening by implementing evidence-based interventions such as patient reminders, provider reminders, provider assessment and feedback, patient navigation, and reducing structural barriers [19,20]. Unfortunately, although CRC screening has significantly increased among Alaska Native and non-Native adults between 1999–2003 and 2014–2018, these CRC prevention and control efforts across the Alaska Tribal Health System have not yet resulted in reduced mortality or cancer stage migration [1,18]. Additional focus on primary and secondary prevention of CRC may be necessary to realize the beneficial decline in CRC mortality experienced by other US populations [21].

Cervical cancer screening among women in this cohort also significantly declined over the ten years of follow-up. One caveat to this finding is that there was a change in the cervical cancer screening guidelines in 2012, with women aged 30–65 years now only requiring a Pap test every five years, coinciding with updated high-risk human papillomavirus testing, rather than every three years if the previous Pap test was negative [22,23]. However, including women who had a Pap test in the last five years still resulted in significantly lower screening rates among the follow-up cohort as compared with baseline. In contrast, reported breast cancer screening using mammography in this cohort remained stable from baseline to follow-up despite changes to the recommended screening age in national U.S. Preventive Services Task Force guidelines [24]. This study found no specific demographic factors associated with cervical or breast cancer screening uptake.

The Centers for Disease Control and Prevention have reported similar trends in screening prevalence among American Indian and Alaska Native people nationally as among Alaska Native and American Indian participants of the Alaska EARTH follow-up study. From 1999 to 2018, national CRC screening rates among American Indian and Alaska Native people decreased by 44% for men and 22% for women [25]. The same national data reported a 26% decrease in cervical cancer screening and a 7% decrease in breast cancer screening among American Indian and Alaska Native people [25].

In contrast to national data for American Indian and Alaska Native people throughout the US, Alaska Behavioral Risk Factor Surveillance System data for Alaska Native people showed that cancer screening rates either stayed stable or increased since 1999–2018 [1]. Between 1999–2003 and 2014–2018, CRC screening significantly increased among Alaska Native people, with about 65.6% of Alaska Native adults aged 50–75 years reporting having been screened for CRC. In fact, the screening rate almost doubled between those time periods for Alaska Native people [1]. During 2016, 84.3% of Alaska Native women aged 21–65 reported having had a Pap test in the past three years. Cervical cancer screening rates among Alaska Native women are slightly higher than among non-Native women; however, they are not significantly different from 2016 rates [1]. During 2018, 83.4% of Alaska Native women aged 50–74 reported having had a mammogram within the last two years. Estimated breast cancer screening rates among Alaska Native women have remained relatively stable [1]. Of note, the percent of Alaska Native adults who received CRC, cervical cancer, and breast cancer screening varied by Tribal health regions [26].

Many factors contribute to decision-making about cancer screening [11,26]. Some studies have identified that positive screening attitudes and gaining knowledge and reassurance are associated with the decision to undergo screening, whereas negative screening attitudes and not wanting to know whether one has cancer are associated with the decision to forego screening [27,28]. This study did not examine attitudes and beliefs that may have contributed to declining CRC and cervical cancer screening rates, which is an important area for further research.

A strength of the current study is that it is the first examination of the longitudinal cancer screening prevalence of urban-dwelling Alaska Native people living in southcentral Alaska. There are several study limitations. The data were self-reported, with questions requiring recall over multiple years. No administrative medical data were collected to confirm the participant’s self-reported cancer screening. Additionally, the follow-up cohort was small compared to baseline, largely due to a lack of current contact information [14]. Because of the smaller sample, there is therefore limited statistical power to identify associations, and study results may not be representative of the larger Alaska Native population, especially as more women participated than men. Likewise, these results may not be generalizable to other American Indian/Alaska Native populations living outside of southcentral Alaska. To maintain consistency with the earlier EARTH study cancer screening paper, we did not include the use of stool tests in the analysis. Of note, the Alaska Native Medical Center CRC Screening Guidelines (2021) recommend colonoscopy as the primary recommended CRC screening test given the higher CRC rates in the Alaska Native population [20]. These recommendations are followed by all of the regional Tribal health organizations that provide care to Alaska Native people throughout the state. However, as stool-based tests improve and options expand, such as the multi-target stool DNA test, additional study of the screening methods used is warranted.

## 5. Conclusions

The results from this study show a decrease in colorectal and cervical cancer screening among Alaska Native people living in the urban southcentral Alaska region over 10 years of follow-up. This information can help fill important gaps in knowledge about cancer screening and identify ways to improve cancer screening programs among this population.

## Figures and Tables

**Figure 1 ijerph-20-06596-f001:**
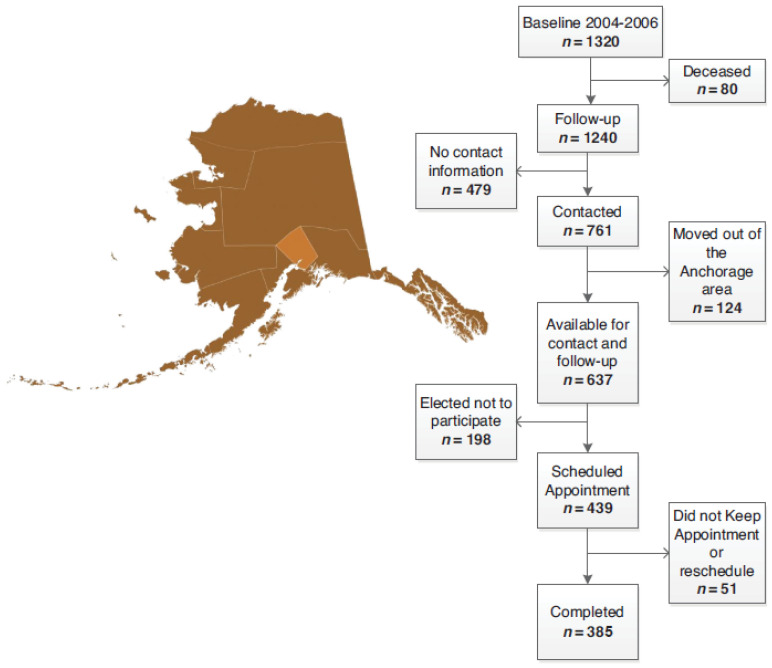
Location of the follow-up Education and Research Towards Health (EARTH) study and participant flowchart.

**Table 1 ijerph-20-06596-t001:** EARTH Study cancer screening questions.

Colonoscopy	Pap Test	Mammogram
Did you ever have a colonoscopy or sigmoidoscopy? These are tests in which a tube is inserted in the rectum to view the bowel.	Did you ever have a Pap smear?	Did you ever have a mammogram?
Yes	Yes	Yes
No	No	No
Not sure	Not sure	Not sure
How old were you when you last had a colonoscopy or sigmoidoscopy?	How old were you when you had your last Pap smear?	How old were you when you had your last mammogram?
___Age Not sure	__Age Not sure	__Age Not sure
If not sure: About how long has it been since you had your last colonoscopy or sigmoidoscopy?	If not sure: About how long has it been since you had your last PAP smear?	If not sure: About how long has it been since you had your last mammogram?
Less than five years ago	Less than five years ago	Less than five years ago
5 to 10 years ago	5 to 10 years ago	5 to 10 years ago
Over 10 years ago	Over 10 years ago	Over 10 years ago
Not sure	Not sure	Not sure

**Table 2 ijerph-20-06596-t002:** Age and characteristics of southcentral EARTH participants at baseline and follow-up study visits.

	Baseline		Follow-Up		*p*-Value
	Mean	SD	Mean	SD	
Age (years)	40.3	12.1	51.9	12.1	<0.01
	** *n* **	**%**	** *n* **	**%**	
**Age (years)**					
18–39	177	46	74	19	<0.01
40+	208	54	311	81	
**Sex**					
Female	279	72	279	72	
Male	106	28	106	28	
**Marital status**					
Married/living as married	175	46	181	47	0.71
**Education level**					
Greater than high school	250	65	260	68	0.14
**Income**					
$40K+	117	32	175	50	<0.01
**Family history of any cancer**					
Yes	141	47	99	41	0.03
**Family history of breast cancer**					
Yes	29	10	45	17	0.02
**Family history of colorectal cancer**					
Yes	48	20	77	32	<0.01
**Cigarette use**					
Ever or Current (smoked at least 100 cigarettes in lifetime)	249	65	251	67	0.08
**Smokeless tobacco use**					
Ever	70	19	59	16	0.08
**Chronic medical conditions**					
No disease or 1 disease	139	36	45	12	
2+ diseases	246	64	333	88	<0.01
**Language at home**					
American Indian/Alaska Native language	55	14	67	18	0.03

**Table 3 ijerph-20-06596-t003:** Proportions of cancer screening test use among southcentral EARTH participants at baseline (*n* = 1320), (2004–2006) and follow-up study visits (385), (2015–2017).

	Baseline		Follow-Up		*p*-Value
	*n*	%	*n*	%	
**Colonoscopy/sigmoidoscopy ^a^**					
Eligible for CRC screening	86	100	148	100	
Yes, within past 5 years	60	70	79	53	0.02 *
Yes, greater than 5 years ago	7	8	42	28	<0.01 *
Never	19	22	27	18	0.59
**Pap test ^b^**					
Eligible for Pap test	244	100	160	100	
Yes, within past 3 years	219	90	116	73	<0.01 *
Yes, greater than 3 years	19	8	36	23	<0.01 *
Never	6	2	8	5	0.28
**Mammogram ^c^**					
Eligible for mammogram screening	150	100	176	100	
Yes, within past 2 years	117	78	136	77	0.98
Yes, greater than 2 years ago	20	13	33	19	0.24
Never	13	9	7	4	0.13

^a^ Colonoscopy/sigmoidoscopy screening restricted to participants 50 years of age or older. ^b^ Pap tests are restricted to women 18 years of age and older who have not had a hysterectomy. ^c^ Mammogram screening is restricted to women 40 years of age and older. * *p*-value of statistical significance.

**Table 4 ijerph-20-06596-t004:** Factors associated with colonoscopy or sigmoidoscopy in the past five years among participants of the southcentral Alaska EARTH study (*n* = 1320), (2004–2006), compared to the southcentral Alaska EARTH Follow-up Study (*n* = 385), (2015–2017).

	Baseline		Odds Ratio	95% CI Lower Upper	*p*-Value	Follow-Up		Odds Ratio	95% CI Lower Upper	*p*-Value
	Yes	%				Yes	%			
**Age (years)**										
50–59	65	76				72	49			
60+	21	24	0.65	0.19–1.9	0.46	76	51	1.6	0.86–3.2	0.13
Total										
**Sex**										
Female	65	76	4.9	1.7–14.3	<0.01	114	77	1.0	0.47–2.2	0.95
**Marital status**										
Married/living as married	52	60	1.5	0.58–3.8	0.41	65	44	1.1	0.6–2.0	0.87
**Education level**										
Greater than high school	59	69	2.0	0.76–5.3	0.16	99	67	1.2	0.58–2.3	0.69
**Family history of any cancer**										
Yes	38	51	0.9	0.34–2.6	0.91	40	42	1.7	0.73–3.9	0.23
**Family history of breast cancer**										
Yes	8	11	3.1	0.5–59.9	0.31	22	21	0.9	0.35–2.4	0.83
**Family history of colorectal cancer**										
Yes	16	24	1.9	0.53–9.2	0.35	28	30	2.8	1.1–7.7	0.03
**Cigarette use**										
Ever or current	57	66	1.1	0.39–2.8	0.91	48	68	1.2	0.61–2.4	0.57
**Smokeless tobacco use**										
Ever	7	8	1.0	0.20–7.3	1.0	21	14	0.8	0.30–1.9	0.57
**Chronic medical conditions**										
2+ diseases	66	77	1.3	0.44–3.8	0.60	139	94	0.9	0.22–3.6	0.89
**Language at home**										
American Indian/Alaska Native language	17	20	0.5	0.18–1.7	0.28	31	21	0.5	0.20–1.04	0.07
**Income**										
$40K+	34	41	1.3	0.48–3.4	0.64	60	43	1.6	0.82–3.2	0.17

Note: Colonoscopy/sigmoidoscopy screening is restricted to participants 50 years of age or older.

**Table 5 ijerph-20-06596-t005:** Factors associated with Pap test screening among participants of the southcentral Alaska EARTH study (*n* = 1320), (2004–2006) compared to the southcentral Alaska EARTH Follow-up Study (*n* = 385), (2015–2017).

	**Baseline**		**Odds Ratio**	**95% CI**	***p*-Value**	**Follow-Up**		**Odds Ratio**	**95% CI**	***p*-Value**
	**Yes**	**%**				**Yes**	**%**			
*** Age (years)**										
18–39	123	50				42	22			
40+	121	50	0.9	0.39–2.1	0.56	133	76	2.7	1.3–5.6	0.01
**Marital status**										
Married/living as married	117	48	1.3	0.54–3.0	0.57	91	52	1.8	0.9–3.7	0.13
**Education level**										
Greater than high school	172	59	1.4	0.54–3.9	0.54	124	78	0.95	0.38–2.2	0.91
**Family history of any cancer**										
Yes	75	43	1.3	0.45–3.9	0.67	60	45	1.1	0.51–2.6	0.74
**Family history of breast cancer**										
Yes	25	13	0.8	0.23–3.5	0.69	29	19	0.5	0.19–1.2	0.11
**Family history of colorectal cancer**										
Yes	32	20	1.3	0.39–5.7	0.72	45	35	1.2	0.52–3.1	0.64
**Cigarette use ****										
Ever or current	89	36	1.4	0.61–3.3	0.40	87	54	0.99	0.48–2.0	0.98
**Smokeless tobacco use**										
Ever	29	12	1.5	NA	0.99	16	10	1.2	0.35–5.6	0.78
**Chronic medical conditions**										
2+ diseases	155	65	1.3	0.53–3.0	0.57	91	52	1.1	0.21–5.1	0.87
**Language at home**										
American Indian/Alaska Native language	35	14	1.6	0.49–4.2	0.41	30	19	0.6	0.25–1.5	0.27
**Income**										
$40K+	76	33	1.1	0.46–3.1	0.78	67	40	1.0	0.49–2.1	0.94

* At baseline, this was restricted to women 18 years of age and older who have not had a hysterectomy. Follow-up was unable to obtain hysterectomy data. ** Cigarette use is defined as ≥100 cigarettes smoked in a lifetime.

**Table 6 ijerph-20-06596-t006:** Factors associated with screening mammograms among participants of the Alaska EARTH study (*n* = 1320), (2004–2006) compared to the southcentral Alaska EARTH Follow-up Study (*n* = 385), (2015–2017).

	Baseline		Odds Ratio	95% CI Lower Upper	*p*-Value	Follow-Up		Odds Ratio	95% CI Lower Upper	*p*-Value
	Yes	%Yes				Yes	%Yes			
*** Age (years)**										
40–49	87	58				54	31			
50+	63	42	3.4	1.4–9.1	<0.01	122	69	1.5	0.70–3.1	0.29
Total	150	100				176	100			
**Marital status**										
Married/living as married	86	58	1.2	0.54–2.6	0.68	83	47	1.1	0.56–2.3	0.73
**Education level**										
Greater than high school	101	68	0.7	0.30-1.7	0.49	136	77	0.5	0.19-1.3	0.19
**Family history of any cancer**										
Yes	67	53	1.0	0.43–2.3	0.98	53	40	1.3	0.57–3.1	0.55
**Family history of breast cancer**										
Yes	16	13	0.5	0.15–1.5	0.16	31	21	0.6	0.27–1.6	0.34
**Family history of colorectal cancer**										
Yes	22	22	1.1	0.39–3.8	0.83	43	33	0.7	0.31–1.9	0.52
**Cigarette use**										
Ever or current	99	66	0.96	0.41–2.1	0.93	112	64	0.7	0.34–1.6	0.44
**Smokeless tobacco use**										
Ever	14	9	1.0	0.29–4.7	0.99	17	10	1.4	0.42–6.2	0.63
**Chronic medical conditions**										
2+ diseases	111	74	1.6	0.67–3.6	0.28	173	98	1.7	0.08–18.4	0.66
**Language at home**										
American Indian/Alaska Native language	25	17	0.5	0.21–1.4	0.19	34	19	2.5	0.93–9.0	0.10
**Income**										
$40K+	51	36	0.9	0.41–2.1	0.83	87	51	1.2	0.56–2.4	0.69

* Mammograms restricted to women aged 40 years and older.

## Data Availability

Data were not available due to tribal research data restrictions. Researchers interested in these data may contact the corresponding author for more information on the process for requesting access to these tribal data.

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
