# Peer review of "Cancer Screening Prevalence among Participants in the Southcentral Alaska Education and Research towards Health (EARTH) Study at Baseline and Follow-Up"

_ijerph, 2023, doi:10.3390/ijerph20166596_

Round 1

Reviewer 1 Report

The manuscript entitled “Cancer screening prevalence among participants in the southcentral Alaska Education and Research Towards Health (EARTH) study at baseline and follow-up” appears to be interesting. The structure of the manuscript appears adequate and well divided in the sub-paragraphs. Moreover, the study is easy to follow, but some issues should be improved. The manuscript needs moderate grammar correction.

1. Title: the title is not appropriate. I suggesting making up it.

2. Paper is replete with grammatical mistakes. Needs rewriting and thorough evaluation.

3. Some references is missed.

4. In order to make the paper more interesting to read, I suggested that the authors could add one graphical abstract to the manuscript.

5. I suggest including clear limitations of the study in the discussion.

Paper is replete with grammatical mistakes. Needs rewriting and thorough evaluation.

Author Response

Response to Reviewers Comments:

Reviewer 1:

Comments and Suggestions for Authors

The manuscript entitled “Cancer screening prevalence among participants in the southcentral Alaska Education and Research Towards Health (EARTH) study at baseline and follow-up” appears to be interesting. The structure of the manuscript appears adequate and well divided in the sub-paragraphs. Moreover, the study is easy to follow, but some issues should be improved. The manuscript needs moderate grammar correction.

1) Title: the title is not appropriate. I suggesting making up it.
Author Response: Our manuscript is about cancer screening rates among Alaska Native people enrolled in the EARTH study in 2004-2006 and then measured again after ten years of follow-up. We tried to make the title as concise as possible while still capturing the relevant details. We would be happy to revise if the editor has additional suggestions for us.

2)  Paper is replete with grammatical mistakes. Needs rewriting and thorough evaluation.
Author Response: We have gone through the paper to remove errors for clarity. See track changes version of the revised manuscript.

3) Some references is missed. 
Author Response: We have added additional references as suggested.

4) In order to make the paper more interesting to read, I suggested that the authors could add one graphical abstract to the manuscript.
Author Response: We are happy to add to the paper but are unfamiliar with graphical abstracts and would appreciate additional clarification on this point.

5) I suggest including clear limitations of the study in the discussion.
Author Response: We have included a more thorough review of the study limitations in the Discussion.

Comments on the Quality of English Language

1) Paper is replete with grammatical mistakes. Needs rewriting and thorough evaluation
Author Response: We have reviewed the paper to remove errors for clarity. See track changes version of the revised manuscript.

Reviewer 2 Report

The authors present the results of longitudinal assessment of selected cancer screening (colorectal, cervical and breast) for Alaska Native and American Indians living in an urban region of southcentral Alaska.

Overall, the study is well-designed, well-conducted and well-presented.  My only comments are as follows:

1) In the Introduction, the authors indicate that screening rates for the listed cancers for the study population are equal to or higher than non-Native Alaskans, yet the Alaska Native peoples experience a disproportionate burden of these cancers.  The manuscript would be strengthened by addressing this discordance with information about the outcomes (e.g., stage distribution, progression-free survival, overall survival).  Has there been a stage shift associated with screening and a concordant improvement in outcome? Why is the incidence of CRC and cervical cancer higher in Alaska Native peoples?

2) Similarly, in the first paragraph of the Discussion, lines 214-217, the authors report that CRC prevention and control efforts have not yet resulted in reduced mortality although CRC screening has significantly increased.  The manuscript would be strengthened by addressing this discordance.  Why?

3) Lastly, the presented data from various sources seems contradictory with regards to cancer screening (decreased vs. stable or increased).  The manuscript would be strengthened by addressing the contradictory data.

Author Response

To Reviewer:
Enclosed is the revised manuscript: “Cancer screening prevalence among participants in the southcentral Alaska Education and Research Towards Health (EARTH) study at baseline and follow-up." We appreciate the review, and think the changes have made this a better manuscript. Please see comments addressed below and included in the track changed version of the revised manuscript.

Thank you for your consideration.

Lauren Smayda, MPH

Response to Reviewers Comments:

Reviewer 2:

Comments and Suggestions for Authors

The authors present the results of longitudinal assessment of selected cancer screening (colorectal, cervical and breast) for Alaska Native and American Indians living in an urban region of southcentral Alaska.

Overall, the study is well-designed, well-conducted and well-presented.  My only comments are as follows:

1) In the Introduction, the authors indicate that screening rates for the listed cancers for the study population are equal to or higher than non-Native Alaskans, yet the Alaska Native peoples experience a disproportionate burden of these cancers.  The manuscript would be strengthened by addressing this discordance with information about the outcomes (e.g., stage distribution, progression-free survival, overall survival).  Has there been a stage shift associated with screening and a concordant improvement in outcome? Why is the incidence of CRC and cervical cancer higher in Alaska Native peoples?
Author Response: Our group has reported in a previous publication that compared 2000-2008 with 2009-2017 data that we observed no difference in CRC incidence and mortality, age at diagnosis, tumor size, tumor location, or stage distribution between the two time periods (Nash et al., 2021). The reasons for the discrepancy between increasing Alaska Native cancer screening rates and the lack of improvement in cancer incidence and mortality (including stage distribution and survival) is not well understood, but may be related to elevated cancer risk factors among Alaska Native people. It may also be that more time needs to accrue before the benefits of screening are seen, as the higher screening rates have only occurred in the past ten years or so. We have added to the first paragraph of the Introduction to address this point.

2) Similarly, in the first paragraph of the Discussion, lines 214-217, the authors report that CRC prevention and control efforts have not yet resulted in reduced mortality although CRC screening has significantly increased.  The manuscript would be strengthened by addressing this discordance.  Why?
Author Response: Our group has reported in a previous publication that compared 2000-2008 with 2009-2017 data that we observed no difference in CRC incidence and mortality, age at diagnosis, tumor size, tumor location, or stage distribution between the two time periods (Nash et al., 2021). The reasons for the discrepancy between increasing Alaska Native cancer screening rates and the lack of improvement in cancer incidence and mortality (including stage distribution and survival) is not well understood, but may be related to elevated cancer risk factors among Alaska Native people. It may also be that more time needs to accrue before the benefits of screening are seen, as the higher screening rates have only occurred in the past ten years or so. We have added to the first paragraph of the Discussion to address this point.

3) Lastly, the presented data from various sources seems contradictory with regards to cancer screening (decreased vs. stable or increased).  The manuscript would be strengthened by addressing the contradictory data.
Author Response: That is correct, there are differences in cancer screening trends between national data and Alaska data as the national data are collected from American Indian and Alaska Native people living throughout the nation whereas Alaska data is specifically Alaska Native people living in Alaska. Of note, the Alaska Native data may not be generalizable to all American Indian and Alaska Native populations living outside of Alaska. We have tried to clarify where the data come from to help the reader understand the screening rates presented. See track changes version of the revised manuscript.  

Reviewer 3 Report

Why authors did just only interviewed patients about colonoscopy regarding colon cancer screening? There are many other screening options that are also valuable.         

A paragraph discussing the study limitations is needed.                                                                                                                                                                                                                                                                                                                                                                                                                                                                                                                                                                                                                                                                                                                                                                                                                                                                                                                                                                                                                                                                                                                                                                                                                                                                                                                                                                                                                                                                                                                                                                                                                                                                                                                                                                                                                                                                                                                                                                                                                                                                                                                                                                                                                                                                                                                                                                                                                                                                                                                                                                                                                                                                                                                                                                                                                                               

Author Response

To Reviewer:
Enclosed is the revised manuscript: “Cancer screening prevalence among participants in the southcentral Alaska Education and Research Towards Health (EARTH) study at baseline and follow-up." We appreciate the review, and think the changes have made this a better manuscript. Please see comments addressed below and included in the track changed version of the revised manuscript.

Thank you for your consideration.
Lauren Smayda, MPH

Response to Reviewers Comments:

Reviewer 3:

Comments and Suggestions for Authors

1) Why authors did just only interviewed patients about colonoscopy regarding colon cancer screening? There are many other screening options that are also valuable.
Author Response: The EARTH study did collect information on fecal occult blood testing in addition to colonoscopy and sigmoidoscopy. However, the original EARTH study cancer screening paper (Schumacher et al., 2007) restricted their analysis to only participants reporting colonoscopy/sigmoidoscopy to focus on cancer detection tests given the high CRC rates in the Alaska Native population. In order to present comparable data, we also restricted our analysis for the current paper. Further exploration of trends in stool-based colorectal cancer screening would be an interesting follow-up study.

2) A paragraph discussing the study limitations is needed.
Author Response: We have included a more thorough review of the study limitations in the last paragraph of the Discussion.

Reviewer 4 Report

Comments

This study assesses a longitudinal cancer screening prevalence of urban-dwelling Alaska Native people living in southcentral Alaska. The authors present baseline (2015-2017) and follow-up (10 years) prevalence of colorectal, cervical, and breast cancer screening among an urban cohort of Alaska Native individuals assessed at baseline and over ten years of follow-up. Of 1,320 original Southcentral Alaska EARTH study participants, 637 had valid contact information and were invited to participate in the follow-up study. Of 385 participants (61%) completed the follow-up study visit.

Although the response was low, as the authors stated, these results provide information for improving cancer screening prevention programs in Alaska Native population. I think the study is a well-described manuscript but has some differences between text and tables that need to be corrected:

Table 2.- Check your numbers because many differ from those in the text.

Table 2.- Change N for n.

Line 128. The authors stated, “Mean age at baseline was 40.3 years and 52.9 years at follow-up.” However, Table 2 showed 40.3 vs 52.9 years.

In Line 130. The authors stated, “The proportion of women increased from 67% at baseline to 72% at follow-up.” However, in Table two the proportion of women at baseline and follow-up is the same (72%).

Line 132.- The authors stated, “Being married/living as married was less prevalent at baseline (39%) compared to follow-up (46%). However, Table two showed 46% vs 47%, respectively.

Line 139.- The authors stated, “Fewer participants reported cigarette use (66% vs 68%). However, Table two showed 65% vs 67%, respectively.

Line 140. The authors stated, “smokeless tobacco use (8% vs. 14%) at baseline compared to follow up.” However, Table 2 showed 19% vs 16%, respectively.

Table3. Change N for n.

Line 181.- Authors wrote “Among the baseline cohort, women ≥50 years were more likely to have had a mammogram than those aged 40–49 years (OR= 3.4; p<0.01). However, in Table 6 the p=0.01. Please check.

Author Response

To Reviewer:
Enclosed is the revised manuscript: “Cancer screening prevalence among participants in the southcentral Alaska Education and Research Towards Health (EARTH) study at baseline and follow-up." We appreciate the review, and think the changes have made this a better manuscript. Please see comments addressed below and included in the track changed version of the revised manuscript.

Thank you for your consideration.
Lauren Smayda, MPH

Response to Reviewers Comments:

Reviewer 4

Comments and Suggestions for Authors

This study assesses a longitudinal cancer screening prevalence of urban-dwelling Alaska Native people living in southcentral Alaska. The authors present baseline (2015-2017) and follow-up (10 years) prevalence of colorectal, cervical, and breast cancer screening among an urban cohort of Alaska Native individuals assessed at baseline and over ten years of follow-up. Of 1,320 original Southcentral Alaska EARTH study participants, 637 had valid contact information and were invited to participate in the follow-up study. Of 385 participants (61%) completed the follow-up study visit.

Although the response was low, as the authors stated, these results provide information for improving cancer screening prevention programs in Alaska Native population. I think the study is a well-described manuscript but has some differences between text and tables that need to be corrected:

1) Table 2.- Check your numbers because many differ from those in the text.
Author Response: We have made corrections to the paper to address the discrepancies between text and table. Please see track changes version of the manuscript for changes.

2) Table 2.- Change N for n.
Author Response: We have changed “N” to “n” in the paper; see track changes in Table 2 and 3.

3) Line 128. The authors stated, “Mean age at baseline was 40.3 years and 52.9 years at follow-up.” However, Table 2 showed 40.3 vs 51.9 years.
Author Response: Table 2 is correct and we have revised the text to match the table.

4) In Line 130. The authors stated, “The proportion of women increased from 67% at baseline to 72% at follow-up.” However, in Table two the proportion of women at baseline and follow-up is the same (72%).
Author Response: Table 2 is correct and we have revised the text to match the table.

5) Line 132.- The authors stated, “Being married/living as married was less prevalent at baseline (39%) compared to follow-up (46%). However, Table two showed 46% vs 47%, respectively.
Author Response: We have corrected the discrepancies between the text and table.

6) Line 139.- The authors stated, “Fewer participants reported cigarette use (66% vs 68%). However, Table two showed 65% vs 67%, respectively.
Author Response:  We have corrected the discrepancies between the text and table.

7) Line 140. The authors stated, “smokeless tobacco use (8% vs. 14%) at baseline compared to follow up.” However, Table 2 showed 19% vs 16%, respectively.
Author Response:  We have corrected the discrepancies between the text and table.

8) Table 3. Change N for n.
Author Response: We have changed “n” to “N” in the paper; see track changes in Table 2 and 3.

9) Line 181.- Authors wrote “Among the baseline cohort, women ≥50 years were more likely to have had a mammogram than those aged 40–49 years (OR= 3.4; p<0.01). However, in Table 6 the p=0.01. Please check.
Author Response: We have corrected Table 6.

Reviewer 5 Report

Overall this is interesting and well-presented work which, as the authors suggest, is a first step in the understanding of cancer screening behavior and uptake in the Alaskan native community.

A few suggestions and comments aimed at strengthening the work:

The introductory paragraph is a little confusing; screening doesn't really 'prevent' cancer (line 30), the word 'screening' is used 3X (lines 31-32), not sure what you mean by 'screen-related cancer' (line 36). Please consider careful revision of this paragraph.

Para 3: Is the southcentral Alaska region 100% urban or are you referring to the actual patient population in your study, i.e. is this an urban subset of the region? Please clarify.

2.2 Study eligibility, line 84: '..defer enrollment for pregnant..'

2.4 Statistical Analysis: I found the concept of 'percentages of paired..' a little confusing. Perhaps this can be deleted?

3.1 Results: Would suggest stating that you state that the follow up cohort represented 29% of the original invited cohort.

Table 2: Suggest moving the top label 'Age (years) to be Right Justified in the Table as the other variables are.

There seems to be an error in the sex distribution data in Table 2.

There are also values in the Table that don't align with the text: Cigarette use and Smokeless tobacco use.

There is no mention as to the possible implications of fecal occult blood testing and its role in screening. Is it possible that endoscopic screening has given way to FOBT and perhaps this explains the lower rates in follow-up. This should be addressed/explored as a possibility in the discussion if there is no available data (ideally you would include this within the screening data) which would be a limitation.

In the last paragraph of the discussion you mention that data is self-reported. As the screening tests are objective data points, presumably they would be captured through administrative data to confirm or bolster the self-reporting. If this can't be done then this should be specified in the limitations. If it can be done, it really should be.

Author Response

To Reviewer:
Enclosed is the revised manuscript: “Cancer screening prevalence among participants in the southcentral Alaska Education and Research Towards Health (EARTH) study at baseline and follow-up." We appreciate the review, and think the changes have made this a better manuscript. Please see comments addressed below and included in the track changed version of the revised manuscript.

Thank you for your consideration.
Lauren Smayda, MPH

Reviewer 5

Comments and Suggestions for Authors

Overall this is interesting and well-presented work which, as the authors suggest, is a first step in the understanding of cancer screening behavior and uptake in the Alaskan native community.

A few suggestions and comments aimed at strengthening the work:

1) The introductory paragraph is a little confusing; screening doesn't really 'prevent' cancer (line 30), the word 'screening' is used 3X (lines 31-32), not sure what you mean by 'screen-related cancer' (line 36). Please consider careful revision of this paragraph.
Author Response: We have revised the first paragraph for clarity.

2) Para 3: Is the southcentral Alaska region 100% urban, or are you referring to the actual patient population in your study, i.e. is this an urban subset of the region? Please clarify.
Author Response: The follow-up cohort included only Alaska Native and American Indian people living in or around southcentral Alaska, Alaska’s largest urban area (Anchorage/Matanuska-Susitna Valley, population ~400,000). This area is an employment and education center for Alaska, and a variety of Tribal groups live in Anchorage, including Athabascan, Inupiaq, Tlingit/Haida/Tsimshian, Unangan (Aleut), and Yup’ik, as well as American Indian tribes from the contiguous United States.

3) 2.2 Study eligibility, line 84: '..defer enrollment for pregnant..'
Author Response: We have revised this paragraph for clarity.

4) 2.4 Statistical Analysis: I found the concept of 'percentages of paired..' a little confusing. Perhaps this can be deleted?
Author Response: We have deleted “percentages of”.

5) 3.1 Results: Would suggest stating that you state that the follow up cohort represented 29% of the original invited cohort.
Author Response: We have added this statement to the paper.

6) Table 2: Suggest moving the top label 'Age (years) to be Right Justified in the Table as the other variables are.
Author Response: We have right justified the label Age (years).  

7) There seems to be an error in the sex distribution data in Table 2.
 Author Response: The table is correct and we have revised the text to match the table.

8) There are also values in the Table that don't align with the text: Cigarette use and Smokeless tobacco use.
Author Response: The table is correct and we have revised the text to match the table.

9) There is no mention as to the possible implications of fecal occult blood testing and its role in screening. Is it possible that endoscopic screening has given way to FOBT and perhaps this explains the lower rates in follow-up. This should be addressed/explored as a possibility in the discussion if there is no available data (ideally you would include this within the screening data) which would be a limitation.
Author Response: The Alaska Native Medical Center CRC Screening Guidelines (2021) recommends colonoscopy as the recommended CRC screening test given the higher CRC rates in the Alaska Native population. These recommendations are followed by all of the regional Tribal health organizations that provide care to Alaska Native people throughout the state. Although there have been some small efforts to expand use of stool based screening for people who can’t get or refuse colonoscopy, in the CRC Control Programs in the Alaska Tribal Health System only about 5-10% of screening is by stool-based tests, with the vast majority of Alaska Native people being screened via colonoscopy. However, as stool-based tests improve and options expand, such as the multi-target stool DNA test, additional study of screening methods used is warranted.

10) In the last paragraph of the discussion you mention that data is self-reported. As the screening tests are objective data points, presumably they would be captured through administrative data to confirm or bolster the self-reporting. If this can't be done then this should be specified in the limitations. If it can be done, it really should be.
Author Response: The study did conduct medical record review to validate 16 chronic conditions (Koller et al. Agreement Between Self-Report and Medical Record Prevalence of 16 Chronic Conditions in the Alaska EARTH Study. Journal of Primary Care & Community Health 2014) which found that self-reported conditions were underreported in relation to the medical record. However, cancer screening was not one of the abstracted conditions. We have added the lack of self-report validation to the limitations section.

Round 2

Reviewer 1 Report

The authors completely responded to the observations made previously. They also inserted useful and clarifying information. Thank you for editing the manuscript. Now, the manuscript is much better than the former.

Author Response

1) The authors completely responded to the observations made previously. They also inserted useful and clarifying information. Thank you for editing the manuscript. Now, the manuscript is much better than the former.
Author Response: Thank you for taking the time to review our paper; we think the changes have made this a better manuscript.

Reviewer 2 Report

The authors have adequately addressed the concerns I raised in my initial review

Author Response

1) The authors have adequately addressed the concerns I raised in my initial review.
Author Response:  Thank you for taking the time to review our paper; we think the changes have made this a better manuscript.

Reviewer 5 Report

Thank you for this revised version. It is much clearer: a few last suggestions:

Line 37: ' ..five year period are as follows; colorectal cancer...'

Line 45: '..people are not well understood and may be related to behavioral and additional..'

Line 91: '..cancer chemotherapy...'

Line 136: '...study visit, comprising 29%.."

Line 150: '..more participants..'

Line 215: '..in 2012 with women aged 30-65 years now only requiring..'

I suggest you add a comment in regard to fecal based screening tests as per your response above into the discussion section on CRC screening.

Tables 4-6 have confusing titles. You should include the terms 'compared to' between naming the 2 cohorts as well as include the total 'n' of each cohort bracketed within the title. Table 6 should specify 'screening mammography'.

Author Response

Thank you for this revised version. It is much clearer: a few last suggestions:
Author Response:  Thank you for taking the time to review our paper, we think the changes have made this a better manuscript.

1) Line 37: ' ..five year period are as follows; colorectal cancer...'
Author Response:  We have fixed this as suggested.

2) Line 45: '..people are not well understood and may be related to behavioral and additional..'
Author Response:  We have fixed this as suggested.

3) Line 91: '..cancer chemotherapy...'
Author Response:  We have fixed this as suggested.

4) Line 136: '...study visit, comprising 29%.."
Author Response:  We have fixed this as suggested.

5) Line 150: '..more participants..'
Author Response:  Unclear of what needs to be changed.

6) Line 215: '..in 2012 with women aged 30-65 years now only requiring..'
Author Response:  We have fixed this as suggested.

7) I suggest you add a comment in regard to fecal based screening tests as per your response above into the discussion section on CRC screening.
Author Response: Thank you for your suggestion; we have added additional information on this point to the limitations section of the discussion.

8) Tables 4-6 have confusing titles. You should include the terms 'compared to' between naming the 2 cohorts as well as include the total 'n' of each cohort bracketed within the title. Table 6 should specify 'screening mammography'.
Author Response:  We have updated Table 4-6 titles with suggested revisions and added the ‘n’ of each cohort bracketed within tables 4-6 titles.